



# Characterizing the surge behaviour and associated ice-dammed lake evolution of the Kyagar Glacier in the Karakoram

Guanyu Li[1,2,3], Mingyang Lv[1,2*], Duncan J. Quincey[4], Liam S. Taylor[4], Xinwu Li[1,2,3], Shiyong Yan[5], Yidan Sun[5], Huadong Guo[1,2,3]

[1]Key Laboratory of Digital Earth Science, Aerospace Information Research Institute, Chinese Academy of Sciences, Beijing 100094, China;
[2]International Research Center of Big Data for Sustainable Development Goals, Beijing 100094, China;
[3]University of Chinese Academy of Sciences, Beijing 100049, China;
[4]School of Geography, University of Leeds, Leeds, LS2 9JT, UK;
[5]Jiangsu Key Laboratory of Resources and Environmental Engineering, School of Environment Science and Spatial Informatics, China University of Mining and Technology, Xuzhou, 221116, China.

*Correspondence to*: Mingyang Lv (lmynju@163.com)

**Abstract.** Glacier surges are prevalent in the Karakoram and occasionally threaten local residents by inundating land and initiating mass movement events. The Kyagar Glacier is well-known for its surge history, and in particular its frequent-blocking

of the downstream valley, leading to a series of high-magnitude glacial lake outburst floods. Although the surge dynamics of the Kyagar Glacier have been broadly described in the literature, there remains an extensive archive of remote sensing observations that have great potential for revealing specific surge characteristics and their relationship with historic lake outburst floods. In this study, we propose a new perspective on quantifying the surging process using successive Digital Elevation Models (DEMs), which could be applied to other sites where glacier surges are known to occur. Advanced

Spaceborne Thermal Emission and Reflection Radiometer DEMs, High Mountain Asia 8-meter DEMs and the Shuttle Radar Topography Mission DEM were used to characterize surface elevation changes throughout the period 2000 to 2021. We also used Landsat time-series imagery to quantify glacier surface velocities and associated lake changes over the course of two surge events between 1989 and 2021. Using these data, we reconstruct the surging process of Kyagar Glacier in unprecedented detail and find a clear signal of surface uplift over the lower glacier tongue, along with uniformly increasing velocities,

associated with the period of surge initiation. Seasonal variations in flow are still evident throughout the surge phase indicating the presence of water at the glacier bed. Surge activity is strongly related to the development and drainage of the terminal ice-dammed lake, which itself is controlled by the drainage system beneath the glacier terminus.

## 1 Introduction

A glacier surge is defined as a quasiperiodic oscillating glacial motion switching between rapid and slow flow, periods that

are known as the active and quiescent phase respectively (Benn and Evans, 2010). Surge-type glaciers account for less than 1% of mountain glaciers, but they are important for studying ice flow instabilities and advancing knowledge of glacier



processes and how they respond to a capricious climate (Benn, 2021; Clarke, 1987; Sevestre and Benn, 2015). Surging glaciers are usually clustered, and are found within areas where inputs and outputs of enthalpy cannot keep balance (Benn et al., 2019). In this context, enthalpy refers to thermal energy and water gained at the glacier bed resulting from friction and geothermal

heat. Sometimes, climate conditions, topographic and geologic features can intensify such imbalances. The Karakoram is one such zone (Guillet et al., 2022; Sevestre and Benn, 2015). In contrast to most glacierised regions, the Karakoram has experienced balanced to slightly positive ice mass change in recent decades (Hewitt, 2005; Gardelle et al., 2013; Farinotti et al., 2020; Hugonnet et al., 2021), and surge activity has been common (Copland et al., 2011; Bhambri et al., 2017; Paul, 2020). Glacier surges and related glacial lake outburst floods (GLOFs) occasionally cause catastrophic damage to local residents and

infrastructure in high mountain regions of Asia (Hewitt and Liu, 2010; Bhambri et al., 2017; Ding et al., 2018; Bazai et al., 2021). Most recently (2018-19) the Shispare Glacier in the central Karakoram surged, forming an ice-dammed lake (Rashid et al., 2020). The lake partially drained in June 2019 and then again in May 2020 affecting the Karakoram Highway and the neighboring village of Hasanabad (Bhambri et al., 2020). Similarly, the surge of Kelayayilake Glacier in the Pamir, which occurred in the spring of 2015, destroyed dozens of herders' huts and inundated valuable grazing meadow (Shangguan et al.,

2016). A better understanding of surge dynamics, how they evolve, and their relationship with GLOFs would therefore be highly informative for local stakeholders.

Although among hundreds of other surge-type glaciers in the Karakoram, Kyagar Glacier has attracted a lot of attention in the literature (Haemmig et al., 2014; Round et al., 2017; Zhang et al., 2020). It is well known for blocking the Shaksgam valley and thus damming the river, which results in the formation of a large ice-dammed lake. It has been a threat over the past two

centuries to more than one million people living in the Yarkant River basin (Yin et al., 2019). GLOFs caused by the Kyagar Glacier have been recorded by two gauging stations, at Kuluklangan and Kaqun, since the 1960s (Zhang, 1992; Haemmig et al., 2014). Attempts to provide an early warning system to alert local communities of an imminent flood have had limited success. Although the early warning station was submerged due to the rapid impoundment of the glacial lake in June, 2015, the system did warn the downstream communities and decision-makers about the coming GLOF (Haemmig et al., 2014).

Previous work has focused on observational data and satellite images to characterize surface velocities and elevation changes over previous surge events (Round et al., 2017) and to discuss their relationship with ice-dammed lake evolution. Round et al. (2017) drew the conclusion that outburst events at the Kyagar Glacier are controlled by basal and subglacial hydrological conditions. They also suggested that the potential for further GLOFs in the period following 2016 was high. Haemmig et al. (2014) and Round et al. (2017) both inferred that previous surge activity had occurred prior to 2000, but did not give direct

evidence of it. Two former surges in the 1970s and 1990s were confirmed by Zhang et al. (2022) indicating an approximate surge cycle of 19-20 years.

Remote sensing images provide an excellent means of characterizing glacier surface conditions, through the calculation of glacier flow fields and mass balance for example. In particular, the long archive of Advanced Spaceborne Thermal Emission and Reflection Radiometer (ASTER) stereo images collected since 2000 can provide an insight into seasonal and annual

changes in glacier surface elevation, especially for surge-type glaciers which experience more dramatic surface changes than





non-surging glaciers. With a surge cycle comparable to the ASTER archive length, the Kyagar Glacier is an ideal case on which to test a new approach for describing glacier surge events using DEMs generated from ASTER images. The primary aim of this study was, therefore, to investigate the use of the ASTER archive to quantify glacier surge elevation change before, during, and after surge events. The High Mountain Asia 8-meter (HMA) DEMs (Shean, 2017a, 2017b) and the Shuttle Radar

Topography Mission (SRTM) DEM (EROS Center, 2018) were added to narrow down uncertainties, with a view to developing a workflow that could be universal and popularized to other surging studies.

To aid discussion of surge dynamics of Kyagar Glacier and GLOF assessment, we used Landsat time-series imagery to generate annual and monthly glacial surface velocity maps from 1989 to 2021 covering two surge events. In addition, we characterized the evolution of the ice-dammed lake up to 2022 and evaluated the potential for outburst floods emanating from this site in the

future.

## 2 Study area

Kyagar Glacier is located on the southern slope of Shaksgam Valley near the boundary of Tarim River basin and Indus River basin (Fig. 1). To the northwest of Kyagar Glacier stands Mount Chogori, known as 'K2'. The glacier is polythermal in nature and the meltwater that is discharged from the Kyagar Glacier supplies the Keleqin River, which is the largest tributary of

Yarkant River. Regional temperatures range from -25 °C to +10 °C and the peak melt season runs from June to September (Zhang et al., 2020). Records from meteorological stations in the neighboring Tibet Plateau and Tarim Basin show an increase of ~2 °C in mean annual temperature since the 1960s (Lv et al., 2020). Precipitation in this region is primarily controlled by the Asian monsoon during summer months, and westerly circulation during winter months (Bolch et al., 2012). Due to its continental nature and high elevation situation, there is relatively little precipitation to nourish the glacier accumulation zone.

Precipitation of just 8.87 mm a$^{-1}$ was recently recorded by the Kyagar automatic monitoring station at an elevation of 4810 m above sea level (a.s.l.) (Wei et al., 2018) (Fig. 2c).

Composed of three tributaries, Kyagar Glacier has a glacierized area of 97.3 km$^2$, flowing from its maximum elevation of 7196 m a.s.l. down to the glacier terminus at 4738 m a.s.l. It is at the glacier terminus where small fluctuations in ice flux can result in a blockage of the Keleqin River in the Shaksgam Valley, and the formation of an ice-dammed lake (Fig.2a). Meltwater from

several other glaciers in the broader catchment also contribute to the lake volume. Glacier centerline elevation profiles (Fig. 2a) show that the western branch is higher than the central branch above the confluence (Fig. 2b). The glacier hypsometry indicates a bottom-heavy morphology (Fig. 2c). The glacier tongue is densely covered by ice crevasses, but there is a general absence of 'classic' surge-type surface features (e.g. looped-moraines) (Copland et al., 2003; Sevestre and Benn, 2015).



## 3 Data and methods

### 3.1 Multi-source remote sensing data

We used three DEM products in addition to Landsat series satellite imagery to facilitate this study (Table 1). DEMs were mostly derived from stereo ASTER imagery and supplemented by the HMA DEM and the SRTM DEM. In total we compiled 85 DEMs presenting elevation conditions from 2000 to 2021. These DEMs were co-registered and differenced to generate elevation change maps over the glacierized area and surrounding stable terrain. Post-processing was conducted to identify the glacier surge period and quantify elevation changes before, during, and after the surge on a pixel level. 373 images acquired by Landsat 5 TM, Landsat 7 ETM+, and Landsat 8 OLI were collected to extract annual and monthly glacial surface velocities and investigate glacial lake evolution between 1989 and 2021.

### 3.1.1 ASTER DEM

The Terra ASTER DEM, also known as the AST14DEM data product (NASA/METI/AIST/Japan Spacesystems and U.S./Japan ASTER Science Team, 2001), is produced by the Land Processes Distributed Active Archive Center (LP DAAC) using two stereo images in the near-infrared (Bands 3N and 3B). AST14DEM images have a spatial resolution of 30 m under the Universal Transverse Mercator (UTM) coordinate system and is more accurate than 25 m root mean square error in all dimensions. The ASTER DEM product was ordered through NASA's Earthdata Search (https://search.earthdata.nasa.gov). In this study, the acquisition dates of 67 ASTER DEMs with less than 75% cloud cover are evenly spread between February 2001 and September 2021 (Table S1). Having a width of 60 km, a single granule was sufficient to cover the entire Kyagar Glacier. Therefore, we used the ASTER DEM as the primary data source for quantifying surface elevation changes.

### 3.1.2 HMA DEM

The HMA DEM dataset is generated from stereoscopic DigitalGlobe satellite imagery released by the National Snow and Ice Data Center (NSIDC) in 2017 (Shean, 2017a, 2017b). Benefiting from the high spatial resolution of predominantly commercial satellites, the HMA DEM has a finer resolution of 8 m than the ASTER DEM. Seventeen HMA DEM strips (with cloud cover lower than 75%) cover the Kyagar Glacier with a temporal coverage from June 2011 to October 2016 (Table S1). Before co-registration, these DEMs were resampled to 30 m and reprojected to the UTM grid as per the ASTER and SRTM DEM. Due to the fewer number than ASTER DEMs, HMA DEMs were taken as the supplementary data for quantifying surface elevation changes (Table S1).

### 3.1.3 SRTM DEM

The Shuttle Radar Topography Mission (SRTM) collected near-global C-band radar data during 11 days in February 2000. Two different radar antennas onboard the shuttle formed a single-pass interferometry, which was used to calculate surface elevation. Released by United States Geological Survey (USGS) in 2015, SRTM 1 arc-second global DEM (version 3.0) has



an accuracy better than 10 m in both horizontal and vertical directions, at least in areas of low relief (Farr et al., 2007). The SRTM DEM was used as reference elevation for co-registration with other DEM datasets. Penetration of the radar signal can be up to several meters in snow- and ice-covered areas leading to uncertainty in the surface elevation retrieval (Rignot et al., 2001), but the impact of this is small when compared to the elevation changes caused by a surge event.

### 3.1.4 Landsat series imagery

As the longest continuous satellite mission observing the Earth, Landsat imagery is perfect for monitoring environment evolution over the past half century. As a consequence of coarse spatial resolution and data scarcity, images acquired by Landsat-1, -2, and -3 with resolution of approximate 60 m would be extremely challenging for feature tracking. Therefore, a total of 373 radiometrically and geometrically corrected images from Landsat 5, 7 and 8 imageries were downloaded from USGS Earth Resources Observation and Science Center (EROS) (https://glovis.usgs.gov/) (Wulder et al., 2022) to investigate

glacial lake area changes. Additionally, feature tracking was applied to all acquired images and the results from 60 pairs with good correlation performance were finally selected to present annual surface displacements from 1989 to 2021 as well as monthly displacements during surge events.

### 3.2 DEM co-registration and outlier filtering

Multi-source DEMs are known to have inconsistent geolocations due to varying data acquisition models, inadequate ground

survey conditions, different data post-processing methods, and other sources (Nuth and Kääb, 2011). Following the workflow in Figure 3, we co-registered the ASTER DEMs and HMA DEMs to the reference SRTM DEM using Nuth and Kääb (2011) and demcoreg (Shean et al., 2016). Demcoreg is a collection of Python and shell scripts for co-registration of DEMs, which automatically implements the correction algorithm by Nuth and Kääb (2011). The mean and standard deviations of DEM offsets over off-glacier areas before and after co-registration are shown in Table S1. After the co-registration, we resampled

all DEMs to the resolution of the coarsest ASTER DEM (30 m) using a cubic convolution interpolation, and then we generated difference maps between ASTER/HMA DEMs and the reference SRTM DEM in order to obtain regional elevation change data.

To remove outliers in each elevation-change map, glacierised and non-glacierised areas were handled separately. Non-glacier pixels with values greater than 3 standard deviations from the mean were discarded (Ragettli et al., 2016), and the vertical

offsets of most selected ASTER/ HMA DEMs relative to SRTM over stable terrain were reduced to within $\pm 2$ m. We assumed elevation changes over the accumulation zone did not exceed 3 m in each year from 2000 to 2021 and therefore filtered pixels above 5600 m a.s.l. with values $\pm 63$ m (Lv et al., 2020). Pixels at elevations below 5600 m a.s.l. were filtered with a threshold of $\pm 150$ m to avoid excluding real values associated with surging. Outlier-free elevation change maps were arranged in chronological order ready for further processing.





### 3.3 Post-processing of DEM differencing data

Using the elevation change data of Kyagar Glacier as a basis, we here present a time series processing method for long-term elevation change monitoring and for the detection of possible surge events.

The Bayesian Estimator of Abrupt change, Seasonality and Trend (BEAST) is a Bayesian model averaging algorithm to decompose time series data into components, including seasonality, trend, and abrupt changes (Zhao et al., 2019). Compared with conventional change point detection algorithms, BEAST does not seek a single best model that may result in poor fitting, but rather combines kinds of models into an ensemble model and fits the seasonal and trend signals based on Bayesian model averaging and Markov Chain Monte Carlo strategies (Cai et al., 2020). Applying the BEAST model to DEM differencing data over Kyagar Glacier pixel by pixel, we fitted trend signals and located periods of abnormal change from 2000 to 2021, on a six-month rolling basis (Fig. 4). Given that BEAST is a general linear regression model using Bayesian probability, we evaluated its performance by common measures such as probability, R-square ($R^2$), and root-mean-square error (RMSE). We also compared the findings from BEAST to the known timing of surge events as recorded in the literature.

We then applied the PieceWise Linear Functions (PWLF) Python package to further decompose the trend in elevation change into multiple piecewise linear function models (Jekel and Venter, 2019). PWLF is based on a differential evolution optimization algorithm, where users can specify the location or numbers of break points (Storn and Price, 1997). For linear function models, the slope of each segment represents the elevation-change rate within the corresponding period, and the breakpoints represent the abrupt time of elevation change resulting from the glacier surge event.

The results of the BEAST process clearly indicated the presence of a surge event between 2010 and 2018. Thus, we limited the detection of break points to this period. After the PWLF process, every pixel over Kyagar Glacier was assigned 6 parameters; elevation-change rates before, during, and after the surge, the dates of surge initiation and termination, and the duration of the surge impact using labels of $k_1$, $k_2$, $k_3$, $T_s$, $T_e$, and $I$ respectively (Fig. 3, Fig. 4i).

### 3.4 Extraction of glacial velocities

In order to measure surface displacements, feature tracking was conducted using COSI-Corr (Leprince et al., 2007). After co-registration, image pairs were cross-correlated using a sliding window between $32 \times 32$ and $8 \times 8$ to measure horizontal displacements between corresponding surface features along the north-south and east-west directions. Glacial velocity ($Vel$) was then calculated through Eq. (1).

$$Vel = \frac{D_y \times \sqrt{P_{n-s}^2 + P_{e-w}^2}}{D_i} \tag{1}$$

Where:

$D_i$ is the period in days between acquisition of each image pair and $D_y$ is assigned 365 which is the number of days in a year.
$P_{n-s}$ and $P_{e-w}$ are the displacements in north-south and east-west directions respectively. Panchromatic bands of Landsat-7 and Landsat-8 with a resolution of 15 m were selected as the potential image pairs between 1999 and 2021. To cover the period





before 1999, band 4 of Landsat-5 was used as features over ice and snow are more recognizable in the near-infrared spectrum. Velocity profiles along centerlines of west and central branches in Figure 2 were extracted and presented in chronological order so as to consider variabilities among different years and during surge events. We manually removed blunders with differences more than 20 m a$^{-1}$ compared to surrounding areas in each velocity profile. It has previously been shown this

method is applicable for velocity analysis of mountain glaciers with varying degrees of debris cover (Lv et al., 2019). Despite the Scan Line Corrector failure in May 2003 (Markham et al., 2004), we found that feature tracking of Landsat-7 image pairs from 2003 to 2013 was still viable for tracking features between the data gaps.

### 3.5 Uncertainty of velocity extraction

We took the residual velocity value over ice-free regions as the uncertainty of the velocity extraction. Landsat sensors have

different spectral features and spatial resolutions leading to different correlation qualities between image pairs. We generated detailed residual value statistics on hillside and valley areas for the three image categories. The mean uncertainties of Landsat 7 and Landsat 8 image correlation results were 1.29 m a$^{-1}$ and 0.69 m a$^{-1}$ respectively, with standard deviations of 5.36 m a$^{-1}$ and 1.01 m a$^{-1}$. Correlation results of Landsat 5 were a little worse having a mean uncertainty of 6.72 m a$^{-1}$ and a standard deviation of 7.12 m a$^{-1}$. Considering that the annual velocities of Kyagar Glacier reach more than 50 m a$^{-1}$ even during the

quiescent phase, we consider those uncertainties acceptable for studying Kyagar Glacier motion characteristics.

## 4 Results

### 4.1 Surging characteristics

Velocity data indicate that the western branch of Kyagar Glacier experienced two surges from 1989 to 2021, and this was validated by visual interpretation. One surge occurred from 1992 to 1998 and the other from 2012 to 2019, confirming the

previously inferred return period of ~20 years (Fig. 5). The first surge peaked in November 1995 following three years of acceleration and declined thereafter to a minimum level over the next three years. The second surge peaked in September 2014 after a two-year acceleration and declined to a minimum over the next five years (Fig. 6).

The two surges both reached a peak monthly velocity of ~800 m a$^{-1}$, 2-6 km from the terminus (Fig. 6a,c). During the quiescent phase (i.e. before 1992 and from 2000 to 2012), the western branch had an average annual velocity of 40-80 m a$^{-1}$ (Fig. 5c,d).

Velocities of the two branches differed above the confluence (after 7 km) and the central part of the tongue also experienced coincident acceleration and deceleration, albeit at a slightly slower rate (Fig. 5a,b,e,f). This implies the surging activity of the western branch did not block the motion of the central branch, rather it promoted ice mass transportation of the central branch to some extent.

Monthly velocity profiles provide more detail of the dynamic evolution of the surge during the active phase. Figure 6d suggests

that seasonal variations in flow are still an important control on velocity rates, peaking during April to October. The two surges both exhibit abrupt acceleration after spring (2014/04-2014/06 in Fig. 6a and 1995/04-1995/11 in Fig. 6c). They reached a



maximum by the end of the same year and started to decline at the beginning of the next year (2014/10-2015/01 in Fig. 6b and 1996/02-1996/04 in Fig. 6c). However, the deceleration was temporally variable; abnormal increases in both 2015/06-2015/07 and 1996/05-1996/09 suggest this was not a single event of summer re-acceleration after the peak velocity in winter.

Kyagar Glacier entered the quiescent phase after 1998. The large volume of ice in the receiving zone rapidly ablated and the reservoir began to recharge, preparing for the next surge (Fig. 7a). During the period 2012 to 2016, the second surge event occurred. The surface elevation of the central branch lowered by 20-30 m in the accumulation zone, evidencing that the surge impact was wider than just on the western branch (Fig. 7b). This is consistent with the velocity profiles, which show that the central branch also experienced surge-like acceleration and deceleration. Although the active phase ended in 2020, part of the

reservoir zone had already started to recharge during the latter part of the surge (Fig. 7c).

The multi-temporal DEM analysis (Section 3.3) revealed that elevation-change rates in the quiescent phase ($k_1$) reached 5 to 7 m a$^{-1}$ in the reservoir and -7 to -9 m a$^{-1}$ in the receiving zone (Fig. 8a). During the surge, change rates ($k_2$) had peak values ranging from -45 to -55 m a$^{-1}$ in the reservoir and 55 to 65 m a$^{-1}$ in the receiving zone (Fig. 8b). In the following few years after the surge impact, change rates ($k_3$) were 4 to 7 m a$^{-1}$ in the reservoir and -3 to -5 m a$^{-1}$ in the receiving zone (Fig. 8c).

Elevation-change rates right after the surge event (Fig. 8c) are lower than those during the entire quiescent phase (Fig. 8a). It is notable that only the lower region (0-4 km from the terminus) and another area (7-10 km from the terminus) exhibited abnormal elevation changes, while changes over the glacier surface 4-7 km from the terminus were small during the surge.

Results of $k_1$, $k_2$, and $k_3$ suggest many pixels over Kyagar Glacier did not experience abrupt elevation changes. Therefore, we only show here the surge initiation ($T_s$) and termination ($T_e$) of the pixels with probabilities greater than 0.5 in the BEAST

process. The earliest $T_s$ was in 2012 detected over part of the receiving zone (Fig. 8d). The latest $T_e$ was in 2017 at the edge of the terminus (Fig. 8e).

## 4.2 Evolution of the ice-dammed lake

We investigated the evolution of the ice-dammed lake using visual interpretation of satellite images and the SRTM DEM (Fig. S1-S11). The lake exhibited an irregular change from 1989 to 2021 (Fig. 9). In the early 1990s, the lake remained stable and

grew slowly. From 1996 to 2009, the lake experienced yearly oscillations both in area and surface elevation. Before 2000, the annual maximum area of the lake exceeded 2 km$^2$ and reached the maximum of 3.21 km$^2$ in September 1998. Within two days of reaching the maximum area, the lake discharged almost entirely, likely resulting in annual GLOFs to the downstream valley. No lake existed between 2009 and 2015 suggesting an open channel developed beneath the glacier. From 2015 to 2021, the lake entered another period of yearly oscillations reaching a maximum area of 2.76 km$^2$ in August 2017. In contrast to the

earlier period, the lake did not fully disappear, and the timing of each reduction in lake area suggests historic lake outbursts usually took place during every summer (July to September).



## 5 Discussion

### 5.1 Surging process of Kyagar Glacier

The semi-automated parameters extracted from successive DEMs have the advantage of being objectively derived and appear
to show a clearer pattern than the elevation change descriptions that former studies have relied on. Combining these values
with the velocity profiles, we could robustly rebuild the surging process during the last decade.

At the beginning of the surge, velocities increased uniformly and slowly along the glacier tongue, with no obvious surge front
forming until April 2014 (Fig. 5e,6a), a pattern which was also recorded by Round et al. (2017). However, the surge front did
not impact the terminus, most likely due to frictional stress at the bed and the disconnected englacial hydrological system
(Björnsson, 1998). There was a zone of intense compression 3-4 km from the terminus resulting in a notable elevation increase.
Consequently, $T_s$ values in 2012 and 2013 were detected over the middle part of the receiving zone (Fig. 8d), while elevations
over the reservoir zone had no obvious change during this period (Fig. 10).

The surge front was clearly evident in the velocity profiles after April 2014 (Fig. 6a). The previous surge in the 1990s also
underwent abrupt acceleration after spring (April 1995) (Fig. 6c). We suggest this could be attributed to two reasons. The first
relates to a reduction in till strength brought about by an increase in water at the bed during spring, as discussed in previous
studies (Cuffey and Paterson, 2010; Round et al., 2017). The second relates to the extrusion stress of the accumulated ice in
the receiving zone during 2012 and 2013, which likely reached the deformation threshold. The abrupt speed-up along the entire
glacier tongue caused the reservoir surface started to rapidly decline in 2014, transporting a massive volume of ice to the
receiving zone (Fig. 10).

Velocities reached a maximum of ~800 m a⁻¹ in October 2014 and declined again in the coming winter and spring (Fig. 6b).
However, velocities increased again in June-August of 2015, which is coincident with the summer speed-up that is evident in
the quiescent phase. Re-acceleration after the peak velocity was also found in May-September of 1996 during the previous
surge (Fig. 6c). We suggest seasonal variations could still play a role in controlling glacial motion during surges. After the
summer of 2015, velocities decreased rapidly to ~ 200 m a⁻¹ within one year and decreased slowly henceforth from 2016 to
2019 (Fig. 5f). In the meantime, a mass wave was generated in the receiving zone propagating towards the terminus (Fig. 8d,e).
We interpret the surge initiation most likely coincided with the maximum volume of ice that could be stored in the reservoir
zone being exceeded, as suggested by Barrand and Murray (2006). The reservoir zone was at capacity at the end of quiescent
phase and transported more ice to downstream than before causing a slow and uniform speed-up from the reservoir to the
receiving zone. Hydrologic processes within and beneath the glacier became important at the same time the surge front was
formed. Once the surge front had passed through the system and the reservoir zone was largely drained, the glacier resumed
its quiescent state, still maintaining a basal hydrology, but with much reduced effect.





## 5.2 Methodological consideration of DEM difference post-processing

The identification of the surge period (2012-2017) using multi-temporal DEM processing is shorter than that when using velocity profiles alone (2012-2019). This can most likely be attributed to the nature of the method. Specifically, we identified
the lattermost profile with a decreasing velocity as the surge termination, even though the reduction may have been small (e.g. <10 m a$^{-1}$). The consequent effect of such velocity changes on the surface elevation would go undetected given the vertical resolution of our data, meaning this period was likely already defined as the start of quiescent phase in PWLF process. The duration of the surge impact ($I$) on individual pixels varied from one to three years (Fig. 8f), meaning that the surge affected most parts of the glacier for discrete periods, rather than for the whole of the active phase.

Uncertainties of elevation change point detection and elevation change rate calculation raised from the processing of BEAST model and PWLF model are shown in Figure 11. For the BEAST model, there exists three parameters to evaluate the uncertainties of the model processing, including the occurrence probability of abnormal change over time, regression evaluation parameters such as $R^2$ and RMSE, as well as the estimated variance of the model error (Zhao et al., 2019). For the PWLF model, various statistics can be calculated if we assume that the breakpoint locations and model form are correct.
Similar to the BEAST model, the PWLF model provides the $R^2$ and standard deviation based on prediction variance to evaluate the uncertainty due to the lack of data in the time series (Jekel and Venter, 2019).

In both models, the output result will be nan/null if the model fitting fails or identified changepoints are not enough. Possible failure reasons include poor data quality, too few data points in time series, or too little variation in the time series. A lower number of null values and highly concentrated valid pixel distributions over the glacier region, indicate a higher reliability of
the model analysis results. As shown in Figure 11, valid pixels are filled in the flatter, lower part of the glacier, while many null pixels exist at high elevation and over steep slopes. This means the model has higher reliability over glacier tongue.

The BEAST and PWLF models are based on general linear regression, so we evaluated the performance of the time series fitting and change point detection by regression parameters including $R^2$ and standard deviation. There are a large number of pixels with high standard deviation and low $R^2$ in the higher part of glacier (Fig. 11a,b,d,e). Due to the BEAST and PWLF
model processing order, $R^2$ in PWLF is higher than that in BEAST, while the standard deviation in PWLF is less than that in BEAST. On the other hand, the processing results of BEAST model contain more information about the data quality and spatial-temporal distribution in time series. However, the coefficient of determination does not necessarily reflect the accuracy of detecting true trend change points (TCPs) in time series. Here, the BEAST algorithm evaluated not only the most likely timings and numbers of TCPs at any given time for each pixel in the glacier, but also the probability of each TCP being true
(Zhao et al., 2019). Compared to the linear regression processing, the detection of TCPs is much stricter. As a result, the pixels with higher $R^2$ and lower standard deviation does not necessarily mean high probability of the TCP being true. In Kyagar Glacier, the pixels with high abnormal change probability ($>= 0.5$) were mainly detected in the reservoir and receiving regions. Although the uncertainty of this approach requires further evaluation, it can quantify surging process in an unprecedented detail. Using this approach, we can investigate elevation-change rates during each period of surge-type glacier, the time of

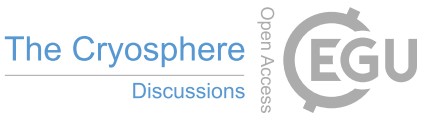

surge initiation and termination, and the surge duration. Compared to manually identifying surges from abrupt changes in DEMs, this approach is much more efficient and objective, only consuming a certain amount of computation load. It would be a valuable attempt to extend the approach to other surge-type glaciers of the whole HMA and even globally, which also makes it possible to identify surge events in a semi-automatic manner based on elevation change data.

### 5.3 Relationship between ice-dammed lake and surges

Previous studies revealed the ice-dammed lake outbursts were related with dynamics of Kyagar Glacier, especially surging events (Haemmig et al., 2014; Zhang et al., 2020). By providing detailed lake evolution data we can offer more comprehensive insights into the relationship between surge and lake outburst events.

We identified two oscillating periods of lake area and surface elevation, which occurred from 1996 to 2009 and 2015 to 2021. Round et al. (2017) also suggested historic lake outbursts could indicate surge activities of Kyagar Glacier. The lake burst
every year during these two periods, although not every GLOF was detected at the gauging station hundreds of kilometers downstream (Chen et al., 2010). This suggests the surge was not the direct reason for triggering the GLOFs. However, we did find a relationship between them, which is that the ice-dammed lakes reached maximum areas after three years when the surge peaked in velocities. For instance, the velocity peaked in 1995 with a maximum lake area of 3.21 km$^2$ in 1998, and in the next surge, velocity peaked in 2014 with a maximum lake area of 2.76 km$^2$ in 2017. We suggest that it would take three years for
the terminus to build a sufficiently large barrier for the lake to develop to its greatest extent after the surge peak.

As a polythermal glacier, it is likely that the nature and efficiency of the hydrological system of the glacier varies greatly on a seasonal basis. Lakes from 1996 to 2009 nearly disappeared every time after outbursts. We suggest that the lake forged an englacial channel through the dam every summer. This interpretation is supported by an evident residual channel in the downstream valley flowing from the terminus area without any obvious changes on the glacier surface (Fig. S1-S6).
Most lake areas from 2015 to 2021 were smaller than during the previous cycle, suggesting the danger associated with the outburst events could be diminishing. During this period, the lake drainage events did not entirely empty the basin (Fig. 9, Fig. S7-S11). This was probably because the connected channel was part way up the ice dam such that the lower part of the lake could not drain. It is therefore likely that it was the alteration of the drainage system beneath (or within) the glacial terminus that controlled the occurrence of each individual GLOF event.

## 6 Conclusion

Using SRTM, ASTER and HMA DEMs, as well as Landsat time-series images, we detected and described the vertical and horizontal glacial motion of Kyagar Glacier and assessed its relationship with GLOFs. By proposing a new quantifying approach based on patterns shown in successive DEMs and combining velocity profiles during last three decades, we constructed the surging process of Kyagar Glacier in an unprecedented detail. As the first aim of this study, (1) we suggest the
new quantifying approach of surging activities basing on the BEAST and PWLF processes could be popularized to other surge-
type glaciers in the HMA or specific regions of Arctic ring. We also find that (2) abnormal uplift over the lower glacier tongue, combined with a uniform increase in velocities, is the clearest indicator of surge initiation. (3) Seasonal variations in flow could still play a role in controlling glacial motion during surges. (4) Surge activities have a strong relationship with the ice-dammed lake evolution, but it is likely to be the alteration of the drainage system within the terminus area that triggered the GLOF events.

*Data availability.* The Landsat series images are available from https://glovis.usgs.gov/ (last access: 10-28-2022). ASTER DEMs can be freely downloaded from https://search.earthdata.nasa.gov (last access: 10-28-2022). SRTM DEM is from USGS EROS Archive. HMA DEMs are provided by NASA NSIDC.

*Supplement.* The supplementary document related to this article is available online at:.

*Author contributions.* ML, DJQ, XL, and HG designed this study. GL, LST, and ML carried out the data processing. ML, GL, and DJQ wrote the manuscript. SY and YS edited every version of this manuscript and assisted in data analyzing.

*Competing interests.* The authors declare that they have no conflict of interest.

*Acknowledgements.* This research is supported by the Strategic Priority Research Program of the Chinese Academy of Sciences (XDA19070202), the National Natural Science Foundation of China (42101124), the Innovative Research Program of the International Research Center of Big Data for Sustainable Development Goals (CBAS2022IRP03), and the International Partnership Program of the Chinese Academy of Sciences (183611KYSB20200059). We greatly thank related organizations and departments for providing data used in this study. We acknowledge the help of Drs. Fanny Brun and Evan Miles for their support on a previous version of this manuscript.

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





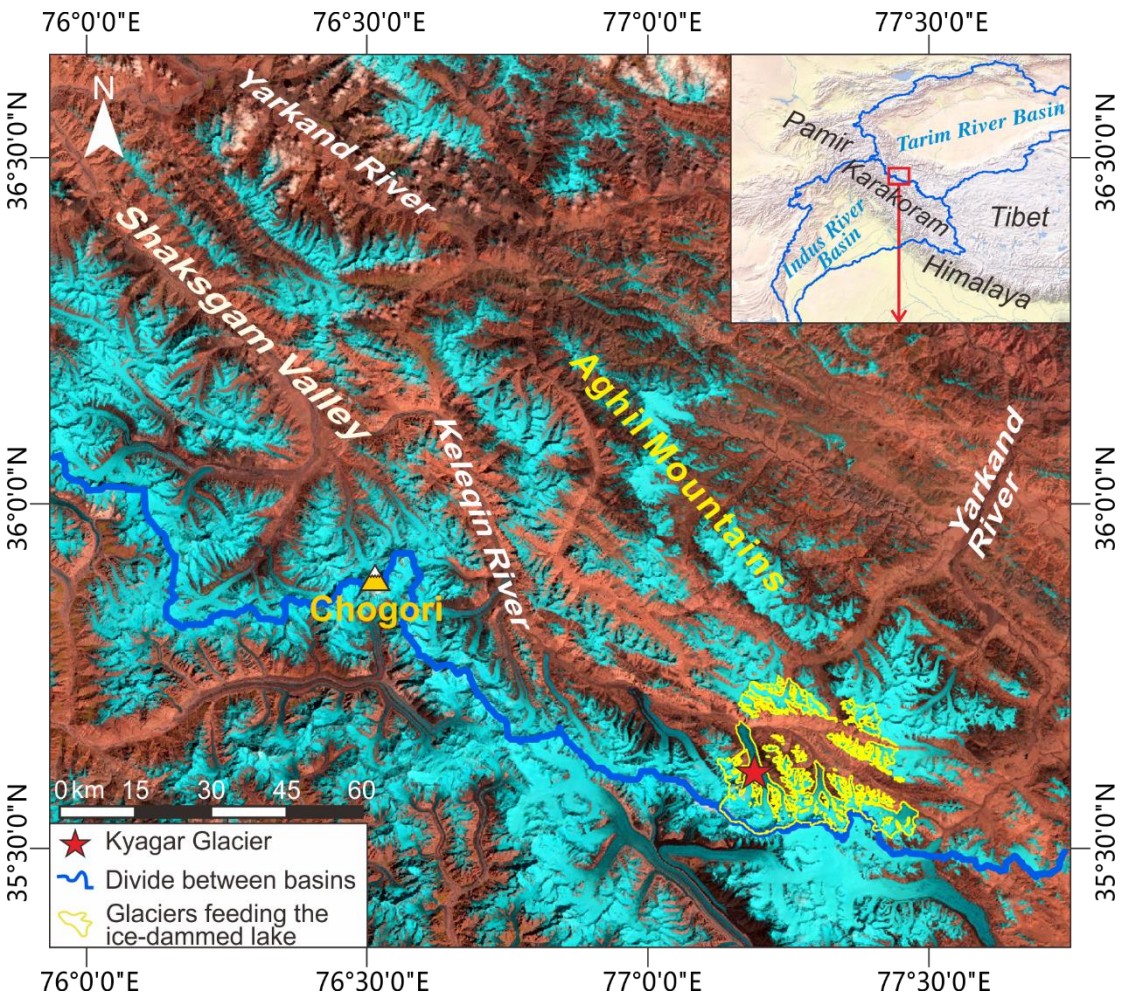

**Figure 1. Location of the Kyagar Glacier and regional geographical conditions. Labelled Mount Chogori is to assist orientation. The background is Landsat OLI image (band 6-5-4) acquired on 6 September 2021.**






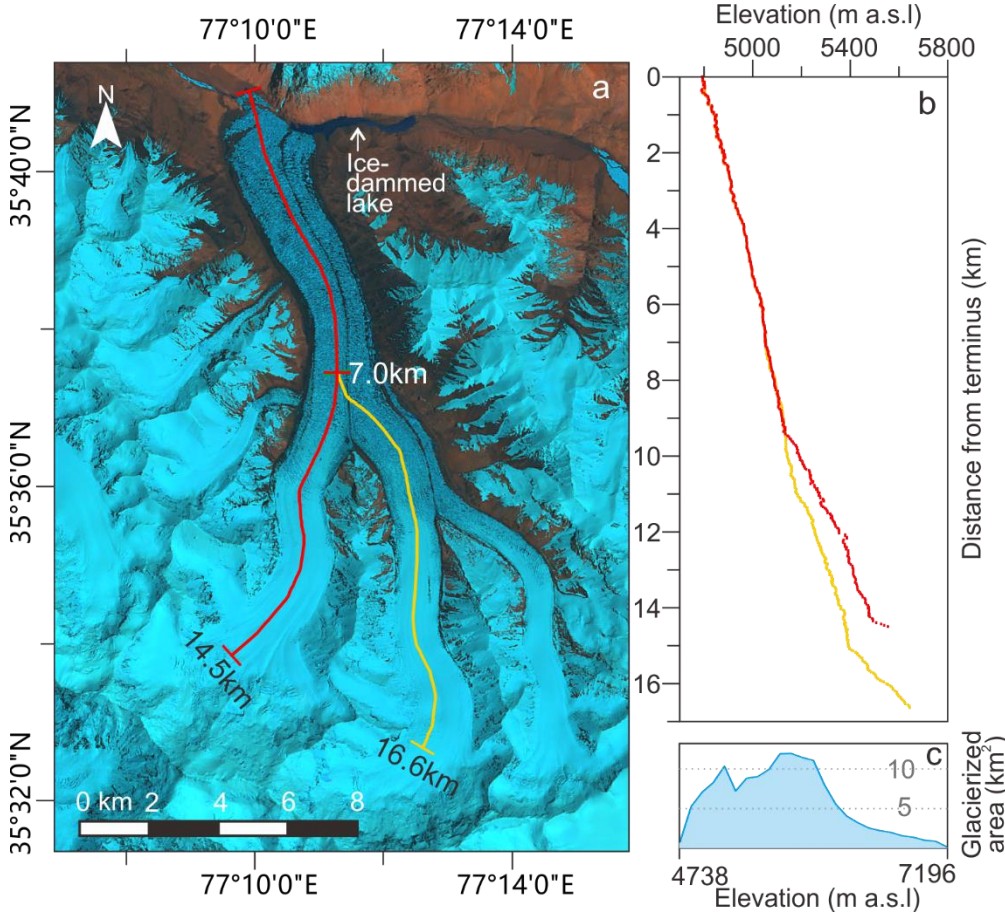

**Figure 2: (a) Location of the ice-dammed lake and elevation profiles of the Kyagar Glacier. The background is a sharpened Landsat OLI image (band 6-5-4) acquired on 4 July 2015. (b) Elevation along two centerlines from the terminus; the western branch is shown in red and the central branch shown in yellow. (c) Hypsometry curve of Kyagar Glacier.**



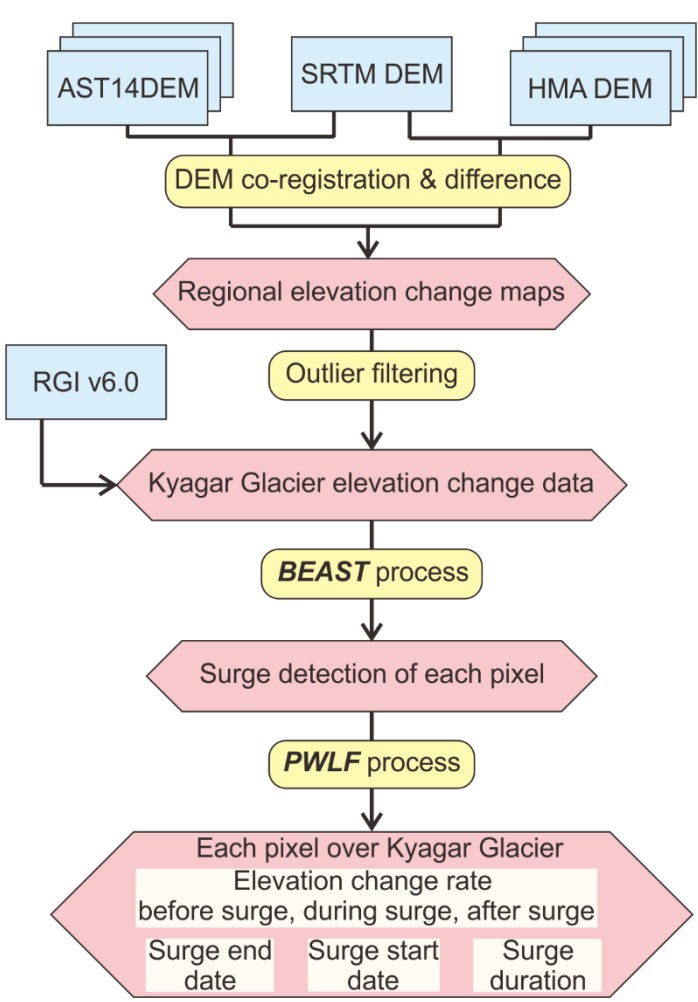

**Figure 3: Workflow to quantify glacial behaviour using multi-temporal DEMs**

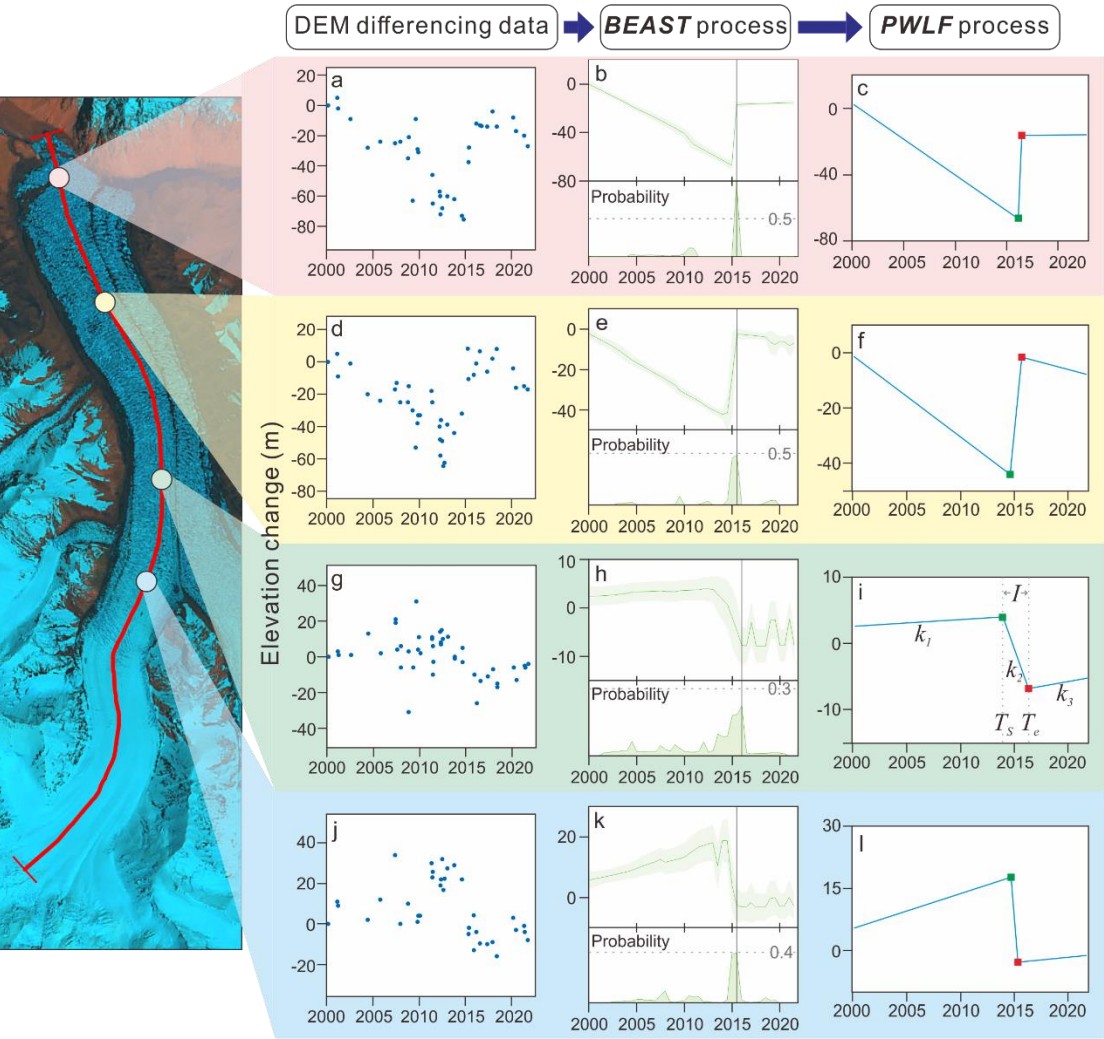

**Figure 4: Presentation of DEM differencing data post-processing. Four random pixels were selected along the central profile of the western branch. The green spots in c, f, i, and l represented the timing of each pixel starting to endure surge impact. The red spots represented the end of surge impact on each pixel. Parameters identified in i are shown in Figure 8.**



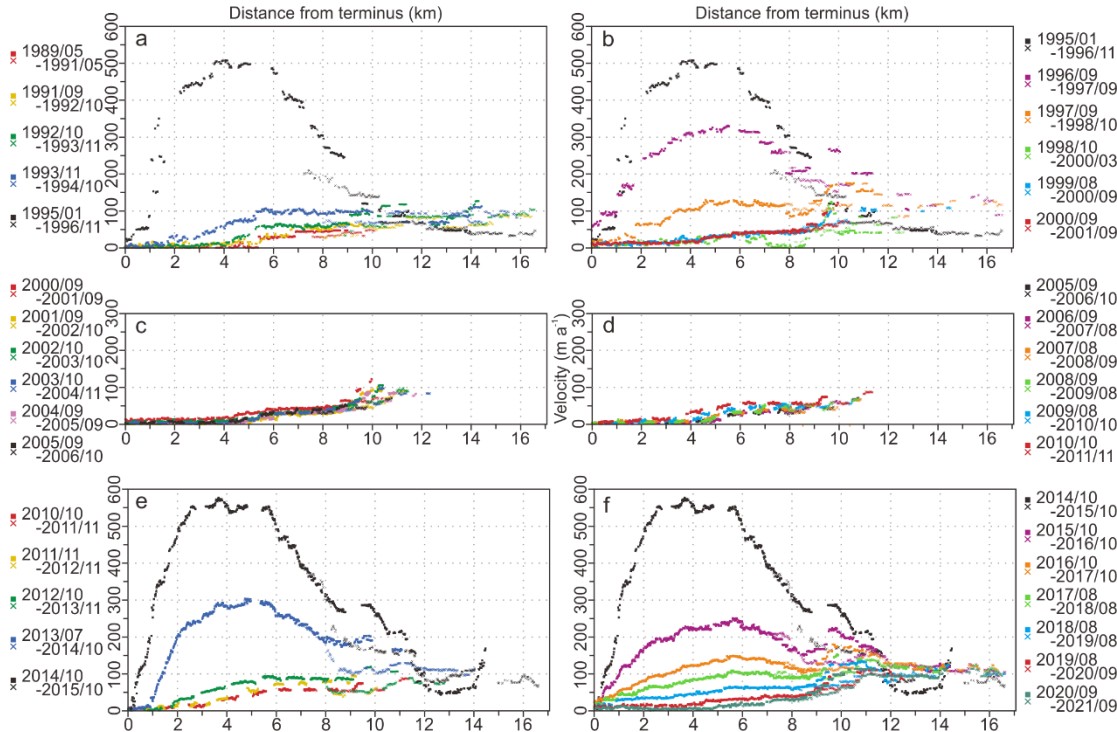

**Figure 5: Annual velocity profiles along the western and central branches of Kyagar Glacier from 1989 to 2021. The centerline of the western branch is formed by square dots and the centerline of the central branch is formed by crosses.**

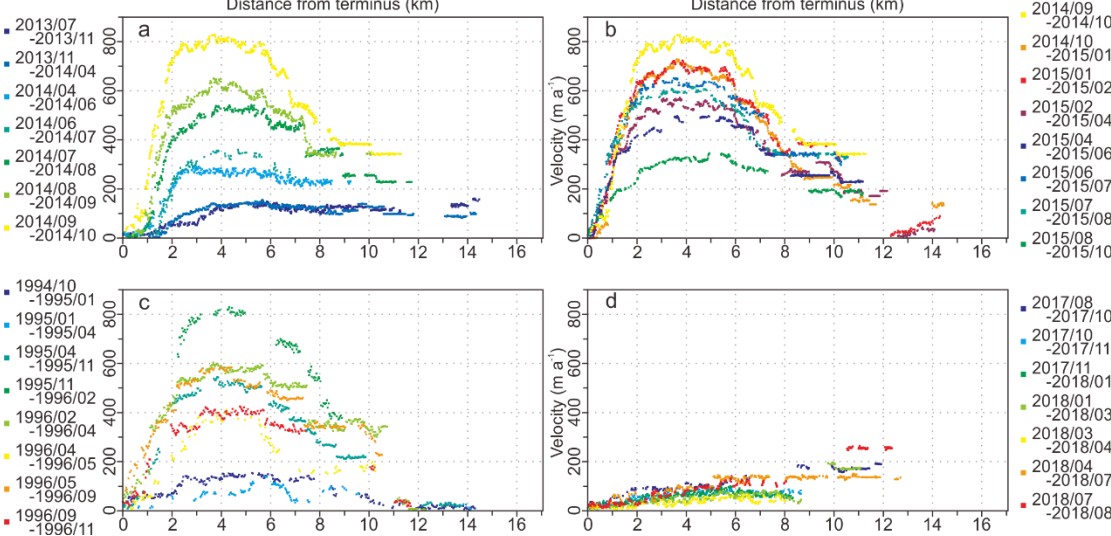

**Figure 6: Near monthly velocity profiles along western branch centerline of Kyagar Glacier during two surging periods.**



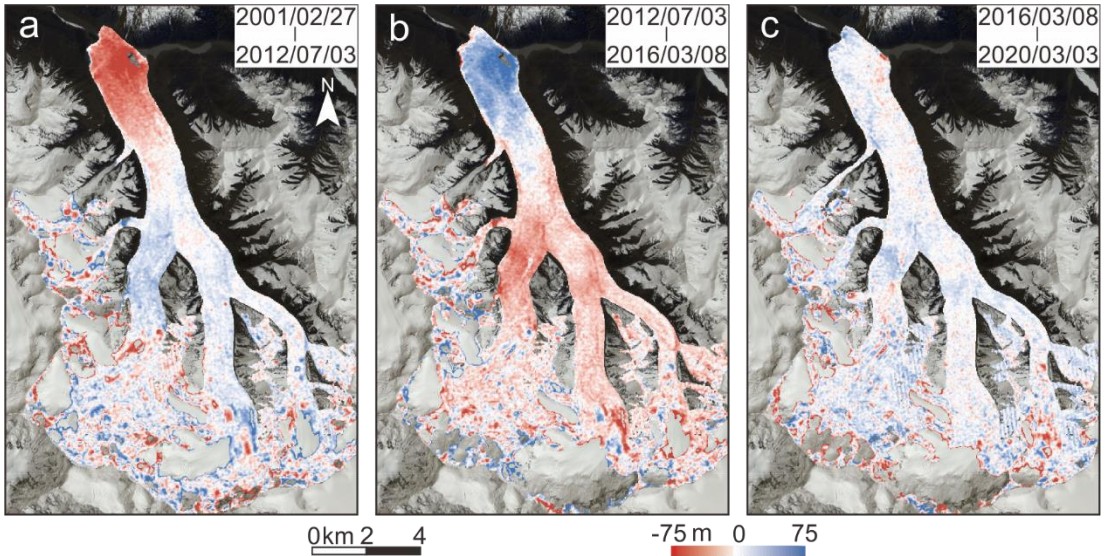


**Figure 7: Surface elevation changes of Kyagar Glacier during three periods (a: 2001 - 2012, b: 2012 - 2016, c: 2016 – 2020).**



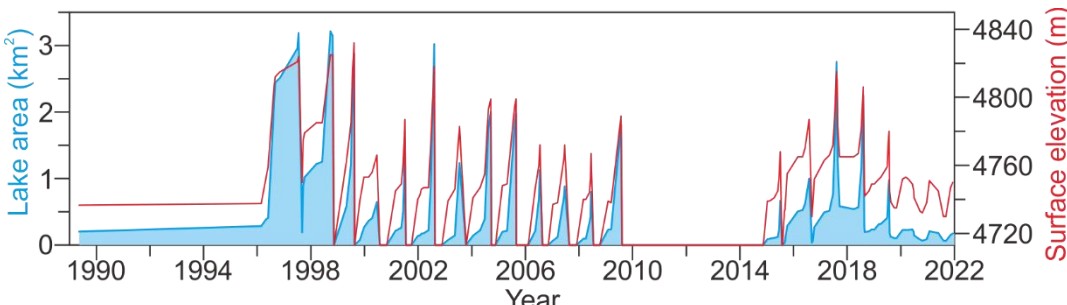

**Figure 8: Parameter maps quantifying elevation changes of Kyagar Glacier. Elevation-change rates before, during, and after the surge of each pixel are labeled as $k_1$, $k_2$, and $k_3$. $T_s$ and $T_e$ represent the dates of surge impact initiated and terminated on each pixel. $I$ represents the duration of surge impact.**

**Figure 9: Evolution of ice-dammed lake surface elevation and area from 1989 to 2021.**


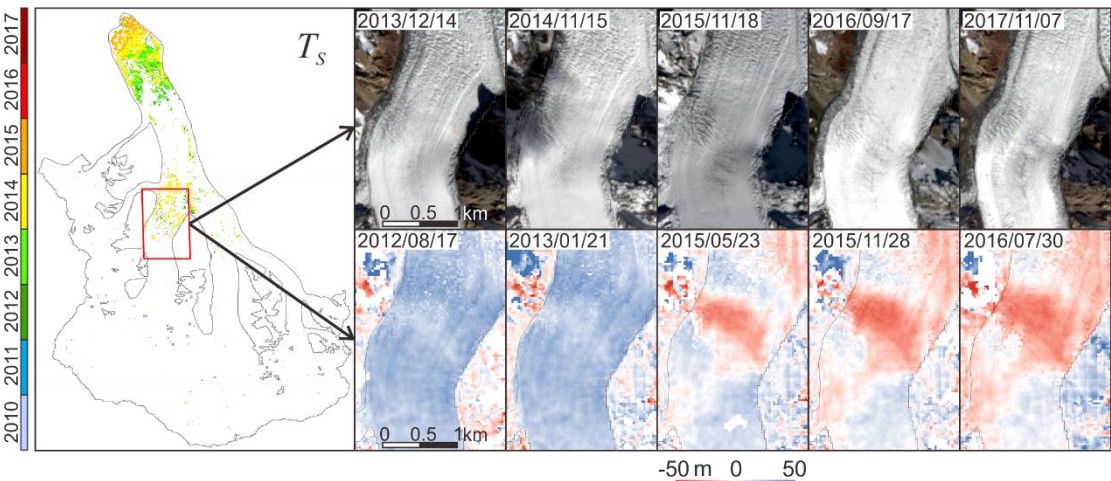

**Figure 10: Details of Landsat 8 OLI images and elevation changes over reservoir zone. Elevation changes between HMA DEM and**
**SRTM DEM are shown.**





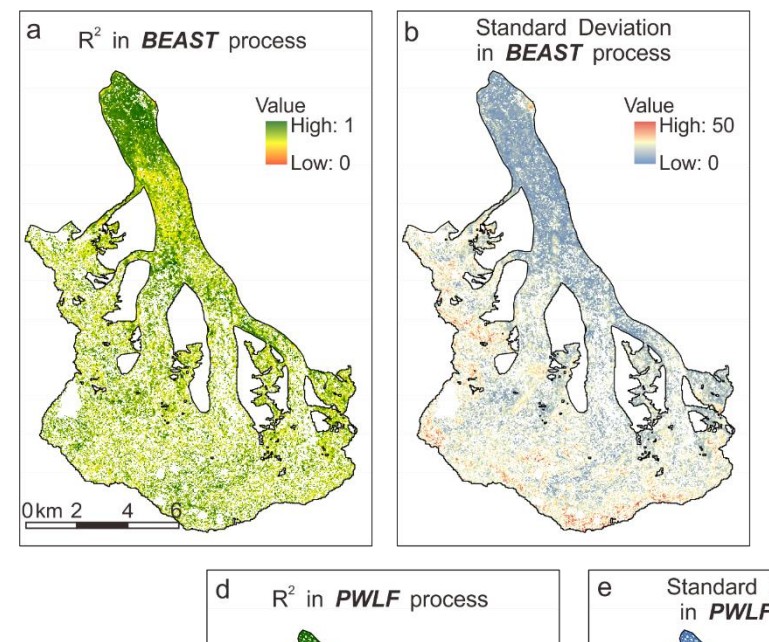

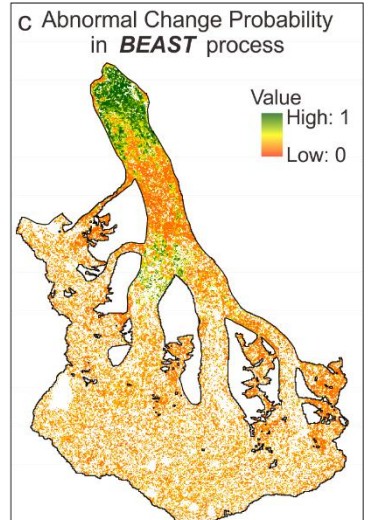

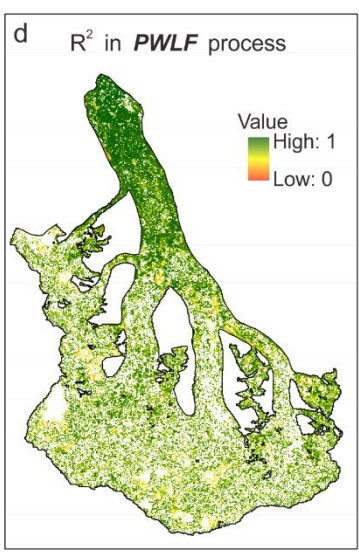

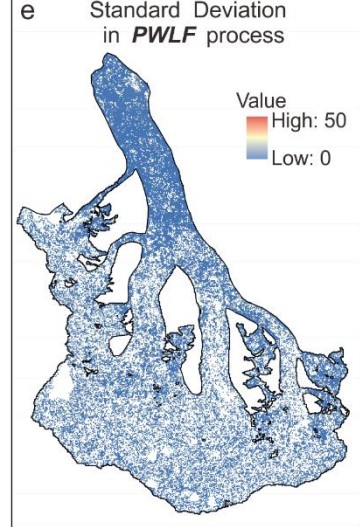

**Figure 11:** $R^2$ **(a), standard deviation (b), and abnormal change probability (c) in BEAST process;** $R^2$ **(d) and standard deviation (e) in PWLF process.**

**Table 1: Details of the Landsat series images and DEMs used in this study**

| Product/Sensor | Resolution (m) | Date of acquisition | Scenes |
|---|---|---|---|
| **DEMs** | | | |
| ASTER DEM | 30 | 2001 Feb 27 – 2021 Sep 30 | 67 |
| HMA DEM | 8 | 2011 Jun 25 – 2016 Oct 2 | 17 |
| SRTM DEM | 30 | 2000 Feb 11-22 | 1 |
| **Landsat imagery** | | | |
| Landsat-5 TM | 30 | 1989 Aug 6 – 2000 Mar 20 | 85 |





| Landsat-7 ETM+ | 30/15(band 8) | 1999 Jul 25 – 2013 May 19 | 147 |
| Landsat-8 OLI | 30/15(band 8) | 2013 Jun 5 – 2021 Dec 11 | 141 |
