# Peer review of "Characterizing the surge behaviour and associated ice-dammed lake evolution of the Kyagar Glacier in the Karakoram"

_The Cryosphere, 2022_

## Author Response (AR1)

**Response to Reviewer #1**

Dear Anonymous Referee #1:

Thank you for reviewing our manuscript. We have carefully considered all your comments and revised the manuscript accordingly. In particular, we appreciate your suggestion to add details about the selection of the reference DEM and filtering method of the dh values. We have reprocessed all the DEM data (1) taking the penetration of SRTM DEM over the snow and ice area into consideration and (2) using new outlier-filtering standard. Also, we have added two new figures depicting annual and monthly glacial velocities of different points along centerline of the western branch.

Here, we respond (in black plain text) to your comments (in blue italics) one by one. We also attach a marked-up version of the revised manuscript and supplementary information.

Best regards,

Mingyang Lv, Guanyu Li and other co-authors

*L54: The GLOF was in 2015. Check the reference. It is from 2014.*

Response: Thanks for this suggestion. The GLOF early warning system was implemented between 2011 and 2013 (hence the reference of Haemmig et al., 2014), and a GLOF occurred in June 2015 while the early warning station was submerged due to the rapid impoundment of the glacial lake (Round et al., 2017). Therefore, we have added another reference (Round et al., 2017) to the sentence here.

*L58: Why was the risk high? More information would be nice*

Response: Thank you for pointing this out. Written in Round et al. (2017), in the period following 2016, 'GLOF hazard potential is expected to remain high for a number of years as the still slightly elevated tongue velocity continues to transport mass to the terminus area, potentially increasing the height of the ice dam until mass transport to the terminus area falls below the ablation rate'. Accordingly, we have added additional information after the sentence in L58. The text now reads 'They also suggested that the potential for further GLOFs in the period following 2016 was high, as the glacier tongue continued to supply mass to the terminus area, maintaining the height of the existing ice dam.'.

*L59: How did they figure it out that there were some surges before 2000?*

Response: Thanks for picking this up, we have added more information to the manuscript.

Haemmig et al. (2014) and Round et al. (2017) found that peak volumes of GLOF in 1978 and 1999 coincide with periods of suspected advance or thickening. They speculated that the clustering of peak flood volumes between late 1990s and early 2000s might indicate a glacier surge prior to 2000. Zhang et al. (2022) found the glacial velocities peaked (surge events) with glacier surface characteristics change in 1975-1978 and 1995-1997 respectively, using historic Landsat imageries and glacial velocity data. Bhambri et al. (2019) also reported one of the surges lasted from 1994 to 1997.

We have changed the original text to 'They speculated that the recurring GLOFs recorded in 1978 and 1999 could indicate the maximum return period of glacier surges. Bhambri et al. (2019) reported one of the surge events lasted from 1994 to 1997. Another former surge from 1975 to 1978 was confirmed by Zhang et al. (2022) using glacial velocity data derived from Landsat images, again indicating an approximate surge cycle of 19-20 years.'.

*L62: replace "surface conditions" with "parameters"*

Response: Thanks for this suggestion. We've changed it accordingly.

*L63: replace "mass balances" with "surface elevation changes"*

Response: Thanks for this suggestion. We've changed it in the new text.

*L78: please provide a reference for "polythermal"*

Response: Thanks for this suggestion. We cited Shi and Liu (2000) in this sentence. According to Shi and Liu (2000), glaciers on the northern slope of mid to western Himalayas and Karakoram are polythermal ones.

*L143: Which areas did you use for the coregistration? All or only off-glacier? Please clarify*

Response: Thank you for pointing this out. We have co-registered the ASTER DEMs and HMA DEMs to the reference SRTM DEM using Demcoreg (Shean et al., 2016). All ASTER DEMs and HMA DEMs covering Kyagar glacier were used for co-registration, including glacier area and off-glacier area. During the process of the Demcoreg software, RGI V6.0 data is used to generate glacier

polygon raster mask for input DEMs, thus technically only off-glacier areas will affect co-registration results.

In the new manuscript, it reads 'Demcoreg is a collection of Python and shell scripts for co-registration of DEMs, which automatically implements the correction algorithm using areas of stable terrain, where the DEMs are least error-prone (Berthier et al., 2016).'.

*L146: Why did you compute the elevation difference relative to SRTM? SRTM is affected by SAR signal penetration. Why don't you use a ASTER DEM from 2000 as a reference? This would reduce the bias. Moreover, SRTM has considerable voids in HMA, filled with other DEM data like ASTER GDEM2. This can lead to some biases in the elevation changes and thus in the subsequent analyses by using e.g. BEAST or PWLF. You should provide information on which areas of the SRTM DEM were filled by other elevation data.*

Response:

Thank you for this suggestion.

1. Suffering from heavy cloud and snow cover, and the rough mountainous topography, individual ASTER DEMs are usually of poor quality and cannot be trusted. ASTER DEMs would not be suitable as a reference for co-registration. Although SRTM is affected by SAR signal penetration, there are many researches and applications using SRTM to calculate glacier mass balance, given its higher precision and well-defined timestamp (Berthier et al., 2016; Lv et al., 2020).

2. Following your suggestion, we are aware that SRTM Non-Void Filled (Digital Object Identifier (DOI) number: /10.5066/F7K072R7) elevation data is more suitable as a reference DEM in our study. As shown in the figure below (Fig. R1), the percentage of void pixels over Kyagar Glacier in SRTM Non-Void Filled data is less than 9%, while most of them are located at the highest elevations. The area below 5600 m affected by the surge contains very few void pixels.

3. Given the known penetration of C-band radar waves over snow and glacier-covered terrain (Gardelle et al, 2013), the SRTM Non-Void Filled data was modified based on the identification of snow- and ice-facies, following previous study in the nearby region. The related text in Section 3.1.3 is now 'In order to correct for surface penetration, we applied adjustments of +5.5 m over firn and snow areas and +1.1 m over clean ice areas following previous studies in the

nearby region (e.g. Lv et al., 2020a). Identification of ice and snow facies is based on a Landsat ETM+ image acquired in February 2000 which was approximately coincident with the SRTM mission'.

[Figure]

**Figure R1: The distribution of void pixels for Kyagar Glacier and its adjacent area in SRTM Non-Void Filled data**

4. Using the resampled non-void filled and penetration-corrected SRTM DEM as the new reference DEM, DEM co-registration, outlier filtering and post-processing of DEM differencing data were totally reprocessed. Parameter maps quantifying elevation changes of Kyagar Glacier in related figures were updated, as well as elevation-change rates data in Section 4.1. Meanwhile, we provide a new version of mean differences and the standard deviation (SD) before and

after co-registration process in the supplement (Table S1). All the updated results are in good agreement with the previous results.

*L152: You should apply a filter that accounts for the time difference relative to your reference DEM. e.g. for a DEM in 2010, the filter should be +/-30 m. Same for the filter for areas below 5600 m.*

Response: Thank you for this suggestion. As you suggested, for the pixels above 5600 m a.s.l., we applied a threshold that varied with time interval of different DEMs. The surface elevation changes below 5600 m a.s.l. could be highly affected by the glacier surge, especially over the terminus area of the Kyagar glacier. Thus, pixels below 5600 m a.s.l. were filtered with a threshold of ±150 m to avoid excluding any real values associated with the surge event. Even though some outliers might still exit after the filtering, they could hardly affect the final result, as the DEM post-processing would probably rule them out.

We used these new outlier-free elevation-change maps for further processing.

*L214ff: It is hard to obtain the temporal evolution from the plot in Figures 5 and 6. It would be helpful if you could provide a plot of the velocity measured at the same spots as for the elevation changes (Fig. 4), or even more spots to better evaluate the temporal evolution of the velocities, in particular the monthly evolution.*

Response: Thank you for this constructive suggestion. According to Figure 5 and 6 (Figure 7 in the new manuscript), we have selected six points along the centerline of west branch (2, 4, 6, 8, 10 ,12 km from the terminus in Fig. R2) to show their annual and monthly velocities (Fig. R3 & R4). These figures and related information have been added to the manuscript.

[Figure]

**Figure R2: The points we selected along the western branch which is 2, 4, 6, 8, 10 ,12 km from the terminus respectively.**

[Figure]

**Figure R3: Annual velocities of different points along the western branch of Kyagar Glacier from 1989 to 2021. The points are selected at 2, 4, 6, 8, 10 ,12 km from the terminus.**

[Figure]

**Figure R4: Near monthly velocities of different points along the western branch of Kyagar Glacier during two surging periods. The points are selected at 2, 4, 6, 8, 10 ,12 km from the terminus.**

*L230: Any explanation why this area did not show any signal?*

Response: Thanks for pointing this out. As shown in Fig. 7 and Fig. 8 (Fig. 9 and Fig. 10 in new manuscript), the elevation change over 4-7 km before and after the surge was much smaller than those of regions over 0-4 km and 7-10 km. Combined with the relatively poor quality of the DEMs we used, especially the ASTER DEMs, very few abnormal change points could be detected by the BEAST and PWLF model over 4-7 km. With better quality DEMs and finer temporal coverage, a clear change signal could also be detected over 4-7 km, even though the signal might be small.

We added a sentence at the end of this paragraph 'This unclear change signal could result from the limited quality of the DEMs at this time'.

*L242: Annual GLOF? You are talking about a single event. If not please provide more information.*

Response: Thank you for this suggestion. We are talking about annual lake outbursts from 1996 to 2009 here. We have changed the original text to 'After reaching its maximum area in each year, the lake discharged almost entirely within about two days, likely resulting in annual GLOFs to the downstream valley from 1996 to 2009'.

*L243: A graph showing the temporal evolution of the lake area (time vs. lake area) would be helpful to illustrate the variations in the lake area.*

Response: We investigated the evolution of the ice-dammed lake using visual interpretation of satellite images and the SRTM DEM (Fig. S1-S11). Meanwhile, we also provide a graph showing the temporal evolution of the lake area and surface elevation in Fig. 9, which is now Fig. 11 in the new manuscript.

*L245: what do you mean by historic? Which period?*

Response: Sorry for the misleading sentence. We have changed the original sentence into two sentences which are now written as 'In contrast to the earlier period, the lake did not fully disappear during each drainage event. The timing of each reduction in lake area suggests historic lake outbursts took place during every summer since 1996 (July to September).'.

*L252: "At the beginning of the surge…." when? Please provide a date information*

Response: Thanks for this suggestion. As shown in Fig. 5e, velocities along the glacier surface 2-4 km from the terminus started to increase slightly since October 2012, which can be considered as the buildup phase before the glacier surge. We have changed the original sentence to 'At the beginning of the surge, velocities increased uniformly and slowly along the glacier tongue since October 2012 (Fig. 5e, 6)' as an independent sentence.

*L254: You mention in the previous sentence that no surge front was formed. And now, you are talking about a surge front. Please clarify*

Response: Sorry for the clumsy expression. We have changed the original sentence "… with no obvious surge front forming until April 2014 (Fig. 5e,6a)" to "It was not until April 2014 that an obvious surge front formed (Fig. 7a), …".

*L255: How did you figure out that there was compression?*

Response: Thanks for this suggestion. For the 3-4 km region, the velocity didn't change obviously from November 2011 to November 2013, while $T_s$ values in 2012 and 2013 were detected. Thus, we suggest that a notable surface elevation increase in 2012 and 2013 over 3-4 km resulted from a compression of ice that could not be transported further downglacier.

In order to clearly express the compression, we have changed the original sentence 'There was a zone of intense compression 3-4 km from the terminus resulting in a notable elevation increase. Consequently, $T_s$ values in 2012 and 2013 were detected over the middle part of the receiving zone (Fig. 8d), while elevations over the reservoir zone had no obvious change during this period

(Fig. 10)' to '$T_s$ values in 2012 and 2013 were detected over the area 3-4 km from the terminus (Fig. 10d), indicating a notable elevation increase, which we speculate resulted from intense compression of ice that could not be transported into the middle part of the receiving zone, while elevations over the reservoir zone had no obvious change during this period (Fig. 12)'.

*L280: which profile? Unclear sentence. Please rephrase this explanation.*

Response: Thanks for pointing this out. We identified the last profile with a decreasing velocity after the peak profile as the surge termination, even though the reduction may be small (e.g.<10 m a$^{-1}$), and this profile is 2019/08-2020/09 profile in Fig. 5f. Now we have changed the original sentence to 'the lattermost profile (2019/08-2020/09 in Fig. 5f)'.

*L302: Why does BEAST contain more information? Please explain*

Response: Thanks for the comment. Compared to the PWLF process, BEAST model was the first step processing the time-series DEMs, which made $R^2$ and standard deviation calculated by BEAST contain more information about raw data quality and spatial-temporal distribution. Now we have changed the explanation to '$R^2$ in PWLF is higher than that in BEAST, while the standard deviation in PWLF is less than that in BEAST. On the other hand, as the BEAST model is firstly conducted to process the time-series DEMs, the results of BEAST could contain more information about raw data quality and spatial-temporal distribution'.

*L307: Please introduce high abnormal change probability and add a cross-link to the respective figure*

Response: Thank you for this suggestion.

1. BEAST model can estimate many parameters of time series data, such as probability of being a changepoint, number of changepoints, probability distribution of total changepoint number (Zhao et al., 2019). BEAST model gives not only a probability distribution of having a changepoint in the trend at each point of time (named as cpOccPr), but also the probabilities associated with the changepoints (named as cpPr). Changepoint is at last identified by the cpPr value, which is calculated by a sum-filtering to the cpOccPr curve. Now we have added the summed cpPr values in the new version Figure 4b,e,h,k. We add the necessary explanation text in Section 3.3 as 'The abnormal changes in an time-series elevation data were identified by the probability

parameters, which are calculated by summing the probability distribution in the trend signals (Fig. 4b,e,h,k)'.

2. The IPCC Likelihood Scale indicates an event is 'more likely than not' to occur when the probability of occurrence is greater than 0.50, 'likely' with probabilities above greater than 0.66, 'very likely' with probabilities above 0.9 and 'virtually certain' when the probability is above 0.99, as shown in Table R1 (Mastrandrea et al., 2010). Thus, we set probability>=0.5 as the threshold to find the changepoints which is more likely than not to occur in this research. We consider that the pixels with probability value large than 0.50 are high enough as possible abnormal change points. Now we have added these information and reference in Section 3.3 as 'According to IPCC Likelihood Scale, the abnormal elevation change is 'more likely than not' to occur when the probability is greater than 0.50 (Mastrandrea et al., 2010). Therefore, we suggest the pixels with probability value larger than 0.50 are credible enough to be considered as abnormal changes'.

**Table R1: Confidence and Likelihood in the IPCC Fifth Assessment Report FACT SHEET**

| Term | Likelihood of the outcome |
|---|---|
| Virtually certain | >99% probability |
| Extremely likely | >95% probability |
| Very likely | >90% probability |
| Likely | >66% probability |
| More likely than not | >50% probability |
| About as likely as not | 33 to 66% probability |
| Unlikely | <33% probability |
| Extremely unlikely | <5% probability |
| Exceptionally unlikely | <1% probability |

*L313: You should also mention that a large amount of DEMs is needed to apply this approach. This might be a limiting factor.*

Response: Thanks for the suggestion. We have added "The only limiting factor for applying this approach would be that a large amount of DEMs with good spatial and temporal coverage are needed." at the end of this paragraph.

**Response to Prof. Rakesh Bhambri**

Dear Prof. Rakesh Bhambri,

Thank you for your time reviewing our manuscript. Your valuable comments are very helpful in improving the content. We have adjusted the manuscript accordingly, including adding some further references and making changes to related sentences and figures as you suggested. In particular, we appreciate your suggestion of adding detailed descriptions of how we extracted the lake area and elevation.

We respond (in black plain text) to all your comments (in blue italics) in-turn, and attach a marked-up version of the revised manuscript and supplement, combined with the changes suggested by our second reviewer.

Best regards,

Mingyang Lv, Guanyu Li, and other co-authors

*This study presented a detailed study of surges of Kyagar Glacier and associated ice-dammed outburst burst floods using a time series of Landsat satellite imagery and ASTER digital elevation models. The manuscript is well-written and nicely structured, and I have given a few minor suggestions for improvement.*

*In the introduction, there is a need to highlight gap areas in previous studies.*

Response: Thanks for the constructive suggestion. We have added necessary information at the end of Paragraph 4 in the Introduction Section to highlight research gaps. It now reads '… Like other glacier surge studies utilizing remote sensing, most of them primarily focused on the evolution of surface velocity and description of related surface features, with limited DEMs used to discuss the amplitude of surface elevation change caused by the surges and analyse mass balance conditions.'.

*L21: 'from' 2000 to 2001.*

Response: Thanks for this suggestion. We've added 'from' here.

*L23: data'sets'.*

Response: Thanks for this suggestion. We've changed accordingly.

*L25: Seasonal variations in 'surface' flow.*

Response: Thanks for this suggestion. We've changed it in the new text.

*L26: Surge activity 'of Kyagar'*

Response: Accepted and changed.

*L29-75: Please refer basic papers on Surge glaciers in the introduction (Jiskoot 2011; Truffer et al. 2021).*

Response: Thanks for listing these helpful references. We have cited Jiskoot (2011) and Truffer et al. (2021) in the first paragraph of the Introduction.

*L35: "The Karakoram is one such zone (Guillet et al., 2022; Sevestre and Benn, 2015)."*
*You can merge a very small sentence with the previous or later sentence.*

Response: Thanks for this suggestion. We have changed the original sentence to "Sometimes, climate conditions, topographic and geologic features can intensify such imbalances, and the Karakoram is one such zone (Guillet et al., 2022; Sevestre and Benn, 2015)."

*L40: You can replace Bhambri et al. (2017) with Bhambri et al. (2019), as the previous reference is related to only the surging process. In contrast, the 2019 reference is based on GLOFs events associated with Surging glaciers.*

Response: Thank you for this suggestion. We've replaced Bhambri et al. (2017) with Bhambri et al. (2019) in this sentence. We still cite Bhambri et al. (2017) in other sentences as we believe it is also supportive to the Introduction Section.

*L56: over previous surge events. You can refer more references (Haemmig et al., 2014; Bhambri et al., 2019).*

Response: Thank you and we have cited Haemmig et al. (2014) and Bhambri et al. (2019) in this sentence.

*L60: Bhambri et al. (2019) also reported a 1990s surge from 1994 to 1997 (Fig 3). You may mention it here.*

Response: Thanks for your suggestion. We have changed the original text to '… They speculated that the recurring GLOFs recorded in 1978 and 1999 could indicate the maximum return period of glacier surges. Bhambri et al. (2019) reported one of the surge events lasted from 1994 to 1997. Another former surge from 1975 to 1978 was confirmed by Zhang et al. (2022) using glacial velocity data derived from Landsat images, again indicating an approximate surge cycle of 19-20 years. …'.

*L66: "Kyagar Glacier is an ideal case on which to test a new approach for describing glacier surge events using DEMs generated from ASTER images". Bhambri et al. (2019) also used time series (10 DEMs) of ASTER DEMs to understand the evolution of surge (Supplementary Fig. S6). However, your study used a large number of ASTER DEMs, and you may highlight this here.*

Response: Thanks for this suggestion. We have cited  Bhambri et al. (2019) in this sentence, and we add extra sentences to highlight the difference of our study at the end of this paragraph. 'As well as using an overall larger number of DEMs to reconstruct the surging process than previous studies (Pitte et al., 2016; Bhambri et al., 2019), we also applied HMA DEMs for the first time to better understand the evolution of the Kyagar Glacier surge.'

*L66: "Kyagar Glacier is an ideal case on which to test a new approach for describing glacier surge events using DEMs generated from ASTER images". You may refer here Pitte et al. (2016), possibly the first used time series of ASTER DEMs for glacier surge study.*

Response: Thank you for mentioning this valuable study. We have cited Pitte et al. (2016) in this sentence.

*L69: the best of our knowledge, no earlier study used HMA DEMs for understanding the evolution of the Kyagar Glacier surge, and you may highlight this here.*

Response: Thanks for this suggestion. We've added this information to the end of this paragraph reading 'As well as using an overall larger number of DEMs to reconstruct the surging process than previous studies (Pitte et al., 2016; Bhambri et al., 2019), we also applied HMA DEMs for the first time to better understand the evolution of the Kyagar Glacier surge.'.

*L83: You may replace Bolch et al. (2012) with Bookhagen and Burbank (2006) and Thayyen, and Gergan (2010).*

Response: Thank you for this suggestion. We've removed Bolch et al. (2012) in the Reference, and add Bookhagen and Burbank (2006) and Thayyen and Gergan (2010) here.

*L93: ice crevasses and "ice pinnacles" (old 1920s field photographs show this).*

Response: Thanks for your suggestion. We've added 'ice pinnacles' in the new text.

*L148-149: "Non-glacier pixels with values greater than 3 standard deviations from the mean were discarded (Ragettli et al., 2016), and the vertical offsets of most selected ASTER/ HMA DEMs relative to SRTM over stable terrain were reduced to within ±2 m." Non-glacier pixels and stable terrain are mentioned in this sentence. Are Non-glacier pixels not containing stable terrain pixels? Please check.*

Response: Sorry for the misleading sentence. The definition of non-glacier pixels and stable terrain are the same in our DEM co-registration and outlier filtering processing. We've changed the 'stable terrain' to 'non-glacierised terrain'.

*L184: Landsat 7. You can shift lines 191-192 on SLC here and link with this sentence.*

Response: Thanks for this suggestion. We've shifted the sentence accordingly. The text now reads 'Panchromatic bands of Landsat-7 and Landsat-8 with a resolution of 15 m were selected as the potential image pairs between 1999 and 2021. Despite the Scan Line Corrector failure in May 2003 (Markham et al., 2004), we found that feature tracking of Landsat-7 image pairs from 2003 to 2013 was still viable for tracking features between the data gaps.'.

*L204: You may replace one with the first surge and the other with the second surge.*

Response: Thanks for this suggestion. We've changed the original sentence to 'The first surge occurred from 1992 to 1998 and the second surge from 2012 to 2019'.

*L238-239: "We investigated the evolution of the ice-dammed lake using visual interpretation of satellite images and the SRTM DEM (Fig. S1-S11)." You can shift this line in the method part.*

Response: Thanks for this suggestion. We add a new Section 3.6 'Visual interpretation of ice-dammed lake evolution' in the method part, and shifted line 238-239 to it. According to the suggestion from Anonymous Referee #1, we replaced original reference 1-arc SRTM data with 3-arc SRTM Non-Void Filled data, which is not good enough for extracting surface elevation of the ice-dammed lake. Therefore, we used HMA DEM on 13 January 2013, when the lake area fully disappeared, as the reference DEM for estimating ice-dammed lake surface elevation. We found that the updated elevation results are in good agreement with our previous results.

The text in Section 3.6 now reads as below:

'We investigated the area of the ice-dammed lake using visual interpretation of satellite images. 148 Landsat images were selected and manually digitised to extract near-monthly lake boundaries (Fig. S1-S11). The HMA DEM obtained on 13 January 2013, when the lake was completely discharged, was taken as reference DEM. We assume the lake bed remained stable during the studied period, and the lake surface elevations at different times were then estimated using the extracted lake boundaries and the reference HMA DEM.'

*Fig 1: Some north-facing glaciers do not contribute to Kaygar glacier ice-dammed lake. Please carefully check.*

Response: Thank you for pointing this out. As shown in the figure below (Fig. R1), The upstream basin of the ice-dammed lake was calculated by ArcGIS hydrology tools. Figure R1 shows that some glaciers in the northern part of the basin have tributaries flowing to the north and south respectively, and the southern tributary contributes to the lake. The glacial boundaries are directly from RGI 6.0, thus we did not manually remove the northern tributaries from the glaciers.

[Figure]

**Figure R1: The upstream basin of Kyagar Glacier ice-dammed lake**

*Fig 2: In this figure, you may mark the western and eastern branches.*

Response: Thanks for this suggestion. We've marked the western and central branches of Kyagar Glacier in the new Figure 2.

*Fig 5 and 6: The legend of this fig may be improved, and image pair information may be present in the same line.*

Response: Thank you for this suggestion. We've improved the legend of related figures to make them clearer to readers.

*Fig 7: You may also add non-glacier elevation change in this fig which will present the accuracy of elevation change at the non-glacier area.*

Response: Thanks for this suggestion. We've added non-glacier elevation change in Figure 7 (Figure 9 in the new manuscript) accordingly.

**Reference**

[revised manuscript text omitted]
_0602_103001000B995100_103001000B62DE00 | 2011/6/25 | -21.73 | 6.068 | 0.216 | 6.30 |
| HMA_DEM8m_AT_20110625_0602_103001000C6D7F00_103001000B48A500 | 2011/6/25 | -18.74 | 6.38 | 0.111 | 5.95 |
| HMA_DEM8m_AT_20120817_0555_103001001ABA5800_103001001BA72400 | 2012/8/17 | -18.02 | 6.30 | 0.107 | 6.05 |
| HMA_DEM8m_AT_20121002_0548_102001001D519C00_102001001E5C8700 | 2012/10/2 | -14.60 | 6.184 | 0.213 | 6.059 |
| HMA_DEM8m_AT_20130121_0537_102001001E3AD500_102001001FD84200 | 2013/1/21 | -17.77 | 6.182 | 0.331 | 6.53 |
| HMA_DEM8m_AT_20150321_0533_104001000970E100_1040010009200000 | 2015/3/21 | -18.94 | 6.08 | 0.23 | 5.67 |
| HMA_DEM8m_AT_20150409_0533_104001000A61D300_104001000A3BBC00 | 2015/4/9 | -17.71 | 6.07 | 0.225 | 5.81 |
| HMA_DEM8m_AT_20150409_0534_104001000A8D7900_104001000955A900 | 2015/4/9 | -17.65 | 5.055 | 0.217 | 5.033 |
| HMA_DEM8m_AT_20150507_0542_1050410012A5C000_1050410012A5BF00 | 2015/5/7 | -19.05 | 6.397 | 0.214 | 6.33 |
| HMA_DEM8m_AT_20150523_0550_1030010042AEB200_103001004177F700 | 2015/5/23 | -23.14 | 6.09 | 0.109 | 5.81 |
| HMA_DEM8m_AT_20151016_0545_1040010012C70E00_104001001380D500 | 2015/10/16 | -16.88 | 6.34 | 0.222 | 5.79 |

| Name | Date | | | | |
|---|---|---|---|---|---|
| HMA_DEM8m_AT_20151128_0742_102 00100481CBF00_10200100470C5800 | 2015/11/28 | -21.61 | 8.16 | -0.07 | 7.56 |
| HMA_DEM8m_AT_20160730_0547_103 0010058028600_103001005A695500 | 2016/7/30 | -19.06 | 6.34 | 0.13 | 6.10 |
| HMA_DEM8m_AT_20161002_0535_105 0010006818700_1050010006818900 | 2016/10/2 | -18.07 | 5.97 | 0.04 | 5.54 |
| HMA_DEM8m_AT_20161002_0534_105 0010006818800_1050010006818600 | 2016/10/2 | -19.26 | 6.36 | 0.23 | 5.79 |
| HMA_DEM8m_CT_20141025_1736_104 0010003AC5200_10300100388CC800 | 2014/10/25 | -12.47 | 6.81 | 0.09 | 6.57 |
| HMA_DEM8m_CT_20141129_1801_103 001003CC3F400_1020010038C4A900 | 2014/11/29 | -18.49 | 6.56 | 0.18 | 6.12 |

**Figure S1.** Sharpened Landsat images showing ice-dammed lake evolution (1).

**Figure S2.** Sharpened Landsat images showing ice-dammed lake evolution (2).

**Figure S3.** Sharpened Landsat images showing ice-dammed lake evolution (3).

**Figure S4.** Sharpened Landsat images showing ice-dammed lake evolution (4).

[Figure]

**Figure S5.** Sharpened Landsat images showing ice-dammed lake evolution (5).

[Figure]

**Figure S6.** Sharpened Landsat images showing ice-dammed lake evolution (6).

**Figure S7.** Sharpened Landsat images showing ice-dammed lake evolution (7).

[Figure]

**Figure S8.** Sharpened Landsat images showing ice-dammed lake evolution (8).

[Figure]

**Figure S9.** Sharpened Landsat images showing ice-dammed lake evolution (9).

**Figure S10.** Sharpened Landsat images showing ice-dammed lake evolution (10).

**Figure S11.** Sharpened Landsat images showing ice-dammed lake evolution (11).

---

## Author Response (AR2)

**Response to Reviewer #1**

Thank you again for your time dealing with our manuscript. We here respond (in black plain text) to your comments (in blue italics). We also attach a marked-up version of the revised manuscript.

Best regards,

Mingyang Lv, Guanyu Li and other co-authors

*Fig 9: Could you please include glacier outlines to better distinguish between the glacier and no-glacier areas? What is the source of the stripes-shaped features next to the glacier?Re-sampling artifacts? Please clarify.*

Response: Thanks for this suggestion. We carefully checked the source of the stripes shaped features and found that it was during the outlier filtering stage, we did not use resampled SRTM DEM to 30 meters as the ASTER DEMs (only during the 1st round reviewing process). This resulted in stripe-shaped features in the DEM differencing data and the stripe features passed on to the previous Figure 9. We corrected this and made a new Figure 9 including glacier outlines.

[Figure]

New Figure 9: Surface elevation changes of Kyagar Glacier during three periods (a: 2001 - 2012, b: 2012 - 2016, c: 2016 – 2020).

*L145: Please cite here also the primary literature: Kääb et al. 2012*

Response: We have cited Kääb et al. 2012 after this sentence.

*L163: Keep Nuth and Kääb 2011*

Response: We have cited Nuth and Kääb 2011 after this sentence.

**Reference**

[revised manuscript text omitted]